# Mediation Effect of Obesity on the Association of Socioeconomic Status with Blood Pressure in the Elderly Hypertensive Population

**DOI:** 10.3390/nu16152401

**Published:** 2024-07-24

**Authors:** Saiyi Wang, Yudong Miao, Yifei Feng, Lipei Zhao, Xiaoman Wu, Shiyu Jia, Weijia Zhao, Clifford Silver Tarimo, Yibo Zuo, Xinghong Guo, Mingze Ma, Jian Wu

**Affiliations:** 1Department of Health Management, College of Public Health, Zhengzhou University, 100 Kexue Road, Gaoxin District, Zhengzhou 450001, China; wangsaiyi001@163.com (S.W.); meldon@zzu.edu.cn (Y.M.); fengyifei2019@163.com (Y.F.); zhaolipei957@163.com (L.Z.); wuxm202311@163.com (X.W.); xk_james@126.com (S.J.); zwjww2009@163.com (W.Z.); 15803828093@163.com (Y.Z.); xinghong_gyo@163.com (X.G.); mamzzu@163.com (M.M.); 2Department of Science and Laboratory Technology, Dar es Salaam Institute of Technology, Dar es Salaam P.O. Box 2958, Tanzania

**Keywords:** socioeconomic status, blood pressure, obesity, mediation effect, elderly hypertensive population

## Abstract

Background: Socioeconomic status (SES) plays a crucial role in blood pressure (BP) control. SES may influence BP control through obesity indices, such as body mass index (BMI) and waist circumference (WC). This study aimed to understand the relationships between SES and BP control in the elderly hypertensive population, and to determine whether BMI and WC mediate the relationship between SES and BP control. Methods: The study was conducted in Jia County, Henan Province, China, from 1 July to 31 August 2023. The 18,963 hypertensive people over 65 years old who were included in the National Basic Public Health Service Program were investigated. The study utilized questionnaire surveys to collect data on participants’ demographic characteristics, disease history, lifestyle behaviors, antihypertensive medication, and measured height, weight, and blood pressure. SES was indexed by participants’ self-reported educational level, family income, and occupation, and categorized into low, medium, and high groups by using latent category analysis (LCA). Logistic regression models were used to analyze the associations between SES and BP control. Obesity indicators, represented by BMI and WC, were included in mediation models to examine the indirect effects of BMI/WC on the association between SES and BP control. Results: The mean age of 17,234 participants was 73.4 years and 9888 (57.4%) of the participants were female. The LCA results indicated the number of participants in low SES, middle SES, and high SES groups were 7760, 8347, and 1127, respectively. Compared with the low SES group, the odds ratios (ORs) and 95% confidence intervals (CIs) for the association of BP control with middle SES and high SES were 1.101 (1.031, 1.175), and 1.492 (1.312, 1.696). This association was similarly found in the subsequent subgroup analyses (*p* < 0.05). Compared with low SES, our findings further suggested that BMI (indirect effects: 95% CIs: −0.004–−0.001; *p* < 0.001) and WC (indirect effects: 95% CIs: −0.003–−0.001; *p* = 0.020) play a suppressing role in the association between high SES and BP control. Conclusions: Our study indicated that the elderly hypertensive population with high SES may have a better result for BP control. However, we found that BMI/WC plays a suppressing role in this association. This indicated that despite the better BP control observed in elderly hypertensive populations with high SES, BMI and WC might undermine this beneficial relationship. Therefore, implementing strategies for obesity prevention is an efficient way to maintain this beneficial association between high SES and BP control.

## 1. Introduction

Socioeconomic status (SES) is a crucial determinant of public and individual health, indicating an individual’s social resources and survival ability [1]. It is a complex index, including race/ethnicity, material resources, income, educational level, and occupation [2]. Currently, many studies often use educational level, income, and occupation as key measures of SES.

Hypertension is the most modifiable risk factor for cardiovascular disease, including stroke, myocardial infarction, and coronary heart disease, which has had a huge impact on the health of the population [3]. Currently, the majority of previous studies have shown a substantial inverse relationship between SES and blood pressure (BP), with a decreased risk among individuals categorized as high SES group [4]. Meanwhile, some investigations also found socioeconomic inequalities have differed between genders [5]. However, most studies have focused on exploring the association of SES and BP in the general population [6,7], while research on this relationship in the elderly hypertensive population remains limited. Therefore, addressing these evidence gaps is of significant public health importance for the elderly hypertensive population and can help alleviate their condition and prevent further complications.

In addition, SES was demonstrated to be strongly associated with obesity in previous studies [8], and research revealed that the prevalence of obesity is shifting towards the most disadvantaged groups from high SES groups as the country develops [9]. A review also indicated that lower educational attainment and income were linked to higher rates of obesity [10]. Meanwhile, obesity was found to be a significant risk factor correlated with high BP [11,12]. Body mass index (BMI) and waist circumference (WC) are commonly used indicators to measure the degree of generalized obesity and central obesity of the human body [13]. Therefore, BMI/WC could be postulated to play a role in the pathway between SES and BP. To date, limited studies have analyzed the effect of obesity on the association of SES with BP control, despite potential implications for long-term human health and well-being.

Despite evidence having shown that there are links between SES with BP, as well as SES and obesity, these relationships have not been thoroughly examined in the elderly hypertensive population. Evidence on how BMI/WC affects SES and BP control in hypertensive populations is still limited. Our study aimed to evaluate the effects of SES, measured through the educational level, family income, and occupation on BP control in the elderly hypertensive population in China, while estimating the role of BMI/WC in this association.

## 2. Methods

### 2.1. Study Population

The study is a countywide, cross-sectional survey of the Chinese hypertensive population, conducted from 1 July to 31 August 2023, in Jia County, Henan Province. The subjects were hypertensive patients aged 65 years or older, enrolled through the National Basic Public Health Service Program. They had been diagnosed with hypertension, agreed to participate in the study, and signed an informed consent form. Subjects with mental illness, secondary hypertension, and those with disabilities or severe disabilities who were unable to complete the questionnaires and physical examinations were excluded. A total of 18,963 hypertensive patients were investigated. The study involved trained research staff to administer questionnaires, collect demographic characteristics, disease history, lifestyle behaviors, antihypertensive medication, and measure height, weight, and BP. We excluded 119 participants with missing data on lifestyle behaviors, 1057 participants with missing data on income data, 105 participants with missing data on educational level, 126 participants with missing data on occupation, 110 participants with missing data on mean diastolic/systolic blood pressure, 101 participants with missing data on BMI and WC, and 111 participants with other information missing. Finally, 17,234 participants were available for current analyses and detailed data information is described in Appendix A. The study obtained ethical approval from the Zhengzhou University Medical Ethics Committee (Number: 2023-318).

### 2.2. Assessment of SES

We used latent class analysis (LCA), using educational level, family income, and occupation to construct a latent variable and estimate SES [14]. Educational level was categorized as illiterate, primary, middle school, and high school or above [15]. Family income was calculated by the interquartile range (income quartile 1: 0–2560 RMB, income quartile 2: 2560–4000 RMB, income quartile 3: 4000–10,000 RMB, income quartile 4: ≥10,000 RMB) [16]. The occupation was categorized as either Farmer or Non-farmer. To select a reasonable latent class number, we used the Akaike information criterion (AIC), Bayesian information criterion (BIC), adjusted BIC (aBIC), Lo–Mendell–Rubins likelihood ratio test (LMR-LRT), and Bootstrap Likelihood Ratio Test (BLRT) for parameter selection. For LMR-LRT and BLRT, results that are not statistically significant suggest that fewer profiles are appropriate. Finally, three latent classes were identified, which respectively represented a high, medium, and low SES according to the item-response probabilities (Appendix A).

### 2.3. Outcome Measurements

In the study, we measured BP levels three times using a standard Omicron electronic sphygmomanometer. Participants were instructed to sit still for five minutes before the first measurement. During the measurements, participants placed their left hand palm up, and a trained primary care worker positioned the armband around the upper arm. The bottom of the armband was positioned 1–2 cm above the inner elbow joint, ensuring it did not cover the joint. The measurement error did not exceed 10 mmHg, and the mean of the three measurements was used to define the systolic and diastolic BP levels. The study subjects were categorized into two groups according to BP levels: controlled and uncontrolled BP. We defined uncontrolled BP as systolic BP ≥ 140 mmHg or diastolic BP ≥ 90 mmHg [17].

### 2.4. Covariate Assessment

Study participants reported their age, gender, marital status, smoking status, drinking status, physical activity, self-reported disease history, and duration of hypertension. Marital status was categorized as accompanied (married) or unaccompanied (unmarried, divorced, widowed). Smoking status was divided into 2 groups: no smoking, which included individuals who had never smoked or had quit smoking for at least 30 years, and smoking, which included individuals who were currently smokers and those who had quit smoking within the past 30 years. Drinking status was grouped into 2 categories: never consumed alcohol and currently drinking. Participants were asked their physical activities in the past 7 days, based on the WHO’s guidelines on physical activity. Healthy physical activity was defined as at least 150 min of moderate intensity or 75 min of vigorous intensity physical activity per week [18]. Participants provided information on their use of antihypertensive medication, including whether they were currently taking it as well as the specific numbers of medications they used.

Participants were instructed to remove any accessories and outer layers of clothing such as hats, coats, or jackets during height and weight measurements. Weight and WC measurements were taken in a fasting state. WC was measured at a level 1 cm above the upper edge of the navel. Each measurement was repeated three times, and the average value was calculated. BMI was calculated as weight in kilograms divided by squared height in meters (kg/m^2^). The guidelines specify four BMI categories: underweight (BMI < 18.5 kg/m^2^), normal weight (18.5 kg/m^2^ ≤ BMI < 24.0 kg/m^2^), overweight (24.0 kg/m^2^ ≤ BMI < 28.0 kg/m^2^), and obese (BMI ≥ 28.0 kg/m^2^); central obesity: male WC ≥ 90 cm, female WC ≥ 85 cm [19]. Both BMI and WC were included as continuous variables in the mediation analyses.

### 2.5. Statistical Analysis

Baseline characteristics of the study sample were reported using frequencies (%) for categorical variables, and continuous variables that were not normally distributed were represented by the interquartile range *P*_50_ (*P*_25_, *P*_75_). Chi-square tests or Kruskal–Wallis H tests were used to determine whether there were statistical differences among groups.

Binomial logistic regression was used to evaluate the association between SES and BP control. Model 1 did not include adjustments for confounding factors, whereas Model 2 controlled for gender, age, marital status, smoking status, drinking status, and physical activity. Model 3 additionally included BMI based on Model 2, while Model 4 added WC based on Model 2. We examined whether the association of BP control and SES might differ across subgroups based on marital status, smoking status, drinking status, BMI levels, and physical activity. In addition, we examined pairwise associations of education level, family income, and occupation to assess their combined impact on BP control.

Our study further conducted the relative mediation effect. The mediating effect of BMI or WC on the relationship between SES and BP control was estimated using the mediation Analysis package in R [20]. SES in the mediation model is a multi-category independent variable. Compared to low SES, a_1_ and a_2_ meant the effects of the middle SES and high SES on BP control, respectively. LCA was conducted using Mplus version 8, and other analyses were run in R software version 4.1.0. All tests were two-sided and *p* < 0.05 indicated statistical significance. And *p* values < 0.05 indicated statistical significance.

## 3. Results

### 3.1. Sample Characteristics

In total, 17,234 participants were included in the analysis, 9888 (57.4%) participants were female, and 9278 (53.8%) had uncontrolled BP. In the stratification of BP control, 39.8% of males and 60.2% of females had uncontrolled BP. A total of 7760 (45.0%) participants were of low SES, 8347 (48.5%) were of middle SES, and 1127 (6.5%) were of high SES. The majority of participants smoked, drank, and had healthy physical activity. The characteristics of participants grouped by their BP control are displayed in Table 1. Univariate analyses showed that gender, age, marital status, drinking status, smoking status, socioeconomic status, major history diseases (coronary heart disease), BMI, WC, use of antihypertensive medication, numbers of antihypertensive medications, and self-reported hypertension duration significantly differed across the controlled BP and uncontrolled BP groups (*p* < 0.05).

In addition, the characteristics of participants by SES were displayed in Appendix A. Participants with low SES were more likely to be female, have a higher prevalence of smoking and drinking, and have higher systolic BP and diastolic BP. Significant differences in gender, age, marital status, drinking status, smoking status, physical activity, history of major diseases (diabetes, stroke, coronary heart disease), BMI, WC, and self-reported duration of hypertension were also observed in SES groups (*p* < 0.05). The characteristics are presented for educational level groups in Appendix A, for income groups in Appendix A, and occupation groups in Appendix A. Among the different education level groups, the highest number of people took one medicine. The same is true for different income groups and occupational groups.

### 3.2. Association between SES and BP Control

As displayed in Table 2, compared with low SES, the higher SES was found to be inversely associated with uncontrolled BP (*P_trend_* < 0.001). The *ORs* and 95% *CIs* for the association of controlled BP with middle SES and high SES were 1.101 (1.031, 1.175) and 1.492 (1.312, 1.696), respectively (Model 4). This association was also observed among individuals who were married, those who did not smoking or consumed no alcohol, and those who had healthy physical activity (Figure 1).

We found that different combinations of educational level, income, and occupation had varied ORs on BP control (Figure 2). Participants with high school or above and those in income quartile 2 tended to have better control of BP. The highest control of BP was observed in individuals with a high school education or above and who were not farmers. Similarly, the highest control of BP was observed in participants in the highest-income quartile who were not farmers.

### 3.3. Effects of BMI and WC on the Association of SES with BP Control

The results of the mediator model analyses are shown in Table 3. Compared with low SES, both BMI and WC did not show a mediating effect on the middle SES with BP control. However, BMIs were observed to partially play a suppressive role in the association between high SES and BP control, as evidenced by the opposite signs of a*b and c. The 95% *CIs* for the total and indirect effects were 0.084 (0.049–0.114) and −0.002 (−0.004–−0.001), respectively. WC had the same function in the high SES, the total and indirect effects were 0.084 (0.054–0.121) and −0.002 (−0.003–−0.001), respectively.

Considering that there was no correlation between SES and BP control in the smoking and drinking groups as shown in Figure 1, we further examined the mediating role of smoking and drinking in the relationship between SES and BP control. Our analysis revealed no mediating effect of smoking and drinking on the relationship between SES and BP control (see Appendix A).

## 4. Discussion

The current study examined the association between SES (operationalized as educational level, income, and occupation) and BP control situation, a known risk factor for poor physical and behavioral health, in the elderly hypertensive population. In addition, we tested the suppressing effect of BMI/WC in this relationship. Our study may provide convincing evidence that SES disparities were associated with differences in BP control among the elderly hypertensive population and call on more targeted prevention measures. These findings are particularly relevant for this population because controlling BP is very important for personal health in the elderly hypertensive population.

Several pieces of evidence have shown that a higher SES can effectively control elevated BP [21]. A study from China found that in plateau regions, lower SES was associated with higher BP levels [22]. A recent systematic review showed that low SES is associated with higher BP [23]. Our findings were consistent with these results, showing that higher SES was inversely associated with high BP compared to low SES. However, a national survey conducted in India found that more than 70% population burden of hypertension was among higher SES groups [24]. Research has shown that in developing countries, the adoption of a more Western lifestyle among high-SES individuals often involves increased consumption of fats, oils, and animal-based foods compared to other population groups [23]. Nevertheless, extensive evidence underscores that low SES is associated with adverse health outcomes, including heightened diabetes risk, increased rates of cardiovascular disease mortality and morbidity, overall mortality, and reduced life expectancy [14,25]. Differences in findings may be attributed to differences in race, study design, and living conditions as well as the level of economic development.

Various studies have demonstrated that socioeconomically disadvantaged groups, particularly those with lower educational levels and poor income, tend to experience more health problems compared to others. A study has shown that a high educational level was associated with a lower risk of BP [26]. Several studies have indicated that disparities in education level contribute to differences in health outcomes including lower knowledge about hypertension and its prevention and control among less educated individuals [27,28,29]. According to previous studies, higher income (income quartile 4) was associated with a lower risk of high BP in this study. Higher income allows people to have access to better medical resources, thus raising the quality of living [30]. In contrast, there were no significant differences observed in BP control across occupation groups. This could be attributed to the focus of our research being on the elderly population, where there may be limited variation in occupations among elderly individuals in rural areas of Henan province. This research further identified how a distinct composition combination of SES may affect the risk of BP control among the elderly hypertensive population in a unique way. Our study found that having a high school education or above and being in the highest income quartile or the second income quartile is associated with better BP control. Additionally, the highest control of BP was observed in individuals with a high school education or above and the highest income quartile who were not farmers. Overall, the research showed that middle SES and higher SES were associated with better BP control for the elderly hypertensive population.

In addition to establishing a link between SES and BP control, our findings indicated that BMI influenced the association between high SES and BP control. Using low SES as a reference, the results suggested that obesity indicators (BMI/WC) partially suppressed the positive effect of high SES on BP control. This implies that obesity is more likely to diminish the beneficial impact of high SES on BP control. Obese conditions may pose a significant challenge in maintaining stable BP [31], general obesity (described as BMI), and abdominal fat (described as WC) and may increase the risk of high BP due to an elevated volume of fatty acid release amplified by insulin resistance and the release of adipokine [32]. The relationship between BMI and BP control may be particularly meaningful in the elderly hypertensive population given links between obesity and high BP-related adverse health concerns. Previous studies have demonstrated that excess adipose tissue in obese hypertensive patients directly triggers chronic overactivation of the sympathetic nervous system (SNS), leading to increased heart rate and vasoconstriction, thereby elevating blood pressure [33]. Additionally, obesity often activates the renin–angiotensin–aldosterone system (RAAS), resulting in heightened levels of renin and angiotensin. These substances promote vasoconstriction and enhance sodium and water retention, further contributing to elevated blood pressure [34]. Reduced levels of adiponectin in obese populations diminish its protective effects on blood vessels, thereby increasing hypertension risk. Furthermore, leptin, a hormone secreted by fat cells, can elevate blood pressure by stimulating the sympathetic nervous system [35]. Our findings suggested a connection between obesity and high SES, which could undermine the beneficial effects of high SES on BP control. Thus, reducing BMI and raising SES might be effective strategies for improving BP control. Additionally, subgroup analysis revealed no statistically significant association between SES and BP control in the smoking and drinking groups. We hypothesized that smoking and drinking might affect the relationship between SES and BP control, prompting us to conduct a mediation analysis. Interestingly, the results indicated that smoking and drinking had no mediating effect. This may be related to the low prevalence of smoking and drinking in our population.

The main strength of this study lies in its large-scale hypertensive population investigation, which provide a certain level of representativeness. Secondly, the examination of three indicators of SES is a notable strength, as it sheds light on the unique relationships between overall SES and each measurement index with BP control. Thirdly, we investigated the relationship between SES and hypertension adjusted for multiple confounders and conducted a stratified analysis with robust results. However, there are several limitations to consider in this study. Given that we examined cross-sectional links between BMI, SES, and BP control, causality could not be determined. Another limitation of this study is that although we used both BMI and WC as a measure of obesity, it lacks specificity in terms of measurement of body fat percentage. Lastly, the self-reported nature of data regarding educational level, income, and occupation introduces the potential for recall bias, thereby affecting the accuracy of these measurements.

## 5. Conclusions

In conclusion, our study found that SES was associated with a risk of high BP. Additionally, obesity indicators, such as BMI and WC, were identified as suppressive factors in the relationship between high SES and BP control. These findings highlighted the importance of not only improving SES but also addressing obesity effectively for better BP control. In addition, more attention should be paid to preventing obesity to avoid the suppression effect on the relationship between high SES and BP control.

## Figures and Tables

**Figure 1 nutrients-16-02401-f001:**
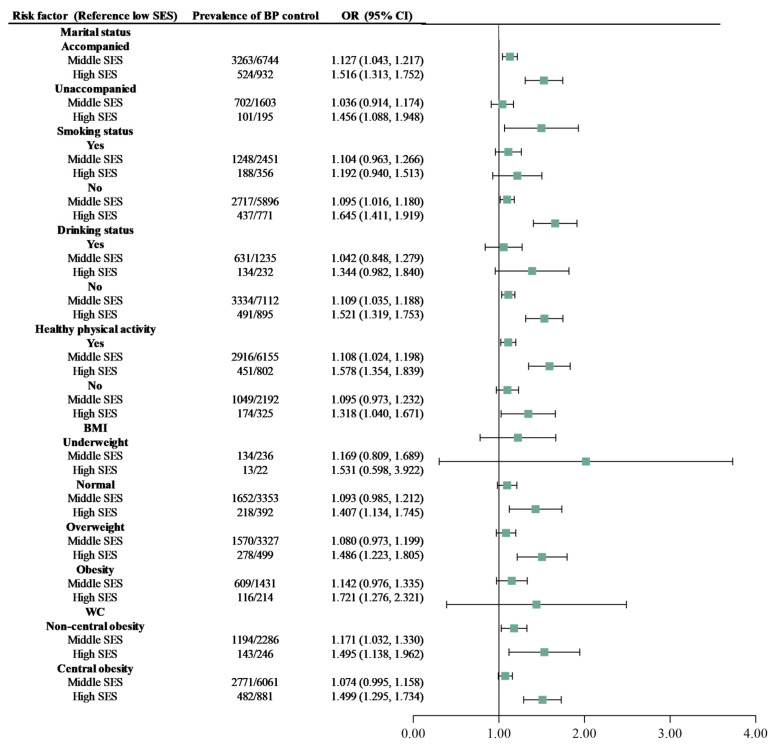
Adjusted odds ratio (OR) of blood pressure control associated with socioeconomic status according to the subpopulation (low SES is the reference group). In addition to adjustment for gender, age, antihypertensive medication-use situation, numbers of antihypertensive medications, hypertension duration, the covariates of marital status, drinking status, smoking status, physical activity, BMI, and WC, were mutually adjusted in the analysis. Small green squares are ORs, and their horizontal lines indicate the corresponding 95% CI. SES: socioeconomic status. BP: blood pressure. BMI: body mass index. WC: waist circumference.

**Figure 2 nutrients-16-02401-f002:**
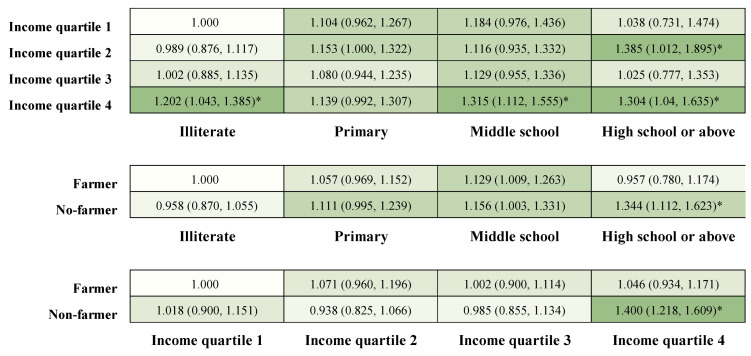
Heatmaps of the associations between education, income, occupation, and blood pressure. Data are OR (95% CI). All models were adjusted for age, gender, marital status, drinking status, smoking status, healthy physical activity, the use of antihypertensive medication, number of antihypertensive medications, duration of hypertension, BMI, and WC. The colors are presented in a gradient, with a darker color shade indicating a stronger association. * *p* < 0.05.

**Table 1 nutrients-16-02401-t001:** Characteristics of participants by BP control.

Variable	Overall (*n* = 17234)	Controlled BP (*n* = 7956)	Uncontrolled BP (*n* = 9278)	*p*
Gender, *n* (%)				<0.001
Male	7346 (42.6)	3654 (45.9)	3692 (39.8)	
Female	9888 (57.4)	4302 (54.1)	5586 (60.2)	
Age, years	73.0 (69.0, 77.0)	72.0 (69.0, 77.0)	73.0 (69.0, 77.0)	<0.001
Marital status, *n* (%)				<0.001
Accompanied	12,393 (71.9)	5866 (73.7)	6527 (70.3)	
Unaccompanied	4841 (28.1)	2090 (26.3)	2751 (29.7)	
Drinking status, *n* (%)				<0.001
Yes	12,998 (75.4)	5850 (73.5)	7148 (77.0)	
No	4236 (24.6)	2106 (26.5)	2130 (23.0)	
Smoking status, *n* (%)				<0.001
Yes	15,180 (88.1)	6903 (86.8)	8277 (89.2)	
No	2054 (11.9)	1053 (13.2)	1001 (10.8)	
Healthy physical activity, *n* (%)				0.867
Yes	12,072 (70.0)	5578 (70.1)	6494 (70.0)	
No	5162 (30.0)	2378 (29.9)	2784 (30.0)	
Socioeconomic status, *n* (%)				<0.001
Low	7760 (45.0)	3366 (42.3)	4394 (47.4)	
Middle	8347 (48.5)	3965 (49.8)	4382 (47.2)	
High	1127 (6.5)	625 (7.9)	502 (5.4)	
Major history diseases, *n* (%)				
Diabetes	4161 (24.1)	1937 (24.3)	2224 (24.0)	0.566
Stroke	5480 (31.8)	2542 (32.0)	2938 (31.7)	0.689
Coronary heart disease	2537 (14.7)	1220 (15.3)	1317 (14.2)	0.035
Use of antihypertensive medication				<0.001
Yes	15,591 (90.5)	7082 (89.0)	8509 (91.7)	
No	1643 (9.5)	874 (11.0)	769 (8.3)	
Numbers of antihypertensive medications				0.005
0	1643 (9.5)	874 (11.0)	769 (8.3)	
1	11,834 (68.7)	5344 (67.2)	6490 (70.0)	
≥2	3757 (21.8)	1738 (21.8)	2019 (21.8)	
Duration of hypertension	10.0 (5.0, 15.0)	10.0 (4.0, 15.0)	10.0 (5.0, 15.0)	<0.001
Body mass index, kg/m^2^	24.6 (22.3,27.1)	24.5 (22.2, 26.8)	24.7 (22.4, 27.2)	<0.001
waist circumference, cm	89.0 (82.0, 95.2)	88.3 (82.0, 95.0)	89.3 (82.5, 96.0)	<0.001
SBP, mmHg	141.0 (129.0, 153.3)	128.0 (120.3, 133.7)	152.0 (145.0, 162.3)	<0.001
DBP, mmHg	77.5 (70.3, 84.7)	72.7 (66.3, 78.6)	82.0 (75.3, 90.0)	<0.001

Notes: Data are reported as a percentage (%) for categorical variables, and the interquartile range *P_50_* (*P_25_*, *P_75_*) for continuous variables that do not conform to the normal distribution. SBP: systolic blood pressure. DBP: diastolic blood pressure.

**Table 2 nutrients-16-02401-t002:** Association of socioeconomic status with BP control.

	Low SES	Middle SES		High SES		
	Ref.	OR (95%CI)	*p*	OR (95%CI)	*p*	*P_trend_*
Model 1	1.000	1.181 (1.110, 1.257)	<0.001	1.625 (1.433, 1.843)	<0.001	<0.001
Model 2	1.000	1.102 (1.033, 1.176)	0.003	1.488 (1.309, 1.692)	<0.001	<0.001
Model 3	1.000	1.100 (1.030, 1.174)	0.004	1.501 (1.320, 1.706)	<0.001	<0.001
Model 4	1.000	1.101 (1.031, 1.175)	0.004	1.492 (1.312, 1.696)	<0.001	<0.001

Notes: Low SES is the reference group. Model 1 did not adjust for confounding factors; Model 2 adjusted for gender, age, marital status, smoking status, drinking status, and healthy physical activity, use of antihypertensive medication, number of antihypertensive medications, and duration of hypertension. Model 3: model 2 and additionally adjusted for BMI. Model 4: model 2 and additionally adjusted for WC. OR: Odds ratio; 95%CI: 95% confidence interval; SES: socioeconomic status.

**Table 3 nutrients-16-02401-t003:** Mediation analysis of BMI and WC on the association between socioeconomic status and BP control.

Mediating Effect Path (Compared with Low SES)	*β*	BootLLCI	BootULCI	*p*	Effect
**Middle SES—BP control (mediating variable: BMI)**					No mediating effect
Total effects	0.008	−0.004	0.025	0.280	
Indirect effects: Middle SES—BMI—BP control	−0.000	−0.001	0.001	0.400	
Direct effects: Middle SES—BP control	0.008	−0.004	0.025	<0.001	
**High SES—BP control (mediating variable: BMI)**					Suppressing effect
Total effects	0.084	0.049	0.114	<0.001	
Indirect effects: High SES—BMI—BP control	−0.002	−0.004	−0.001	<0.001	
Direct effects: High SES—BP control	0.086	0.052	0.116	<0.001	
**Middle SES—BP control (mediating variable: WC)**					No mediating effect
Total effects	0.008	−0.007	0.024	0.360	
Indirect effects: Middle SES—WC—BP control	−0.000	−0.000	0.000	0.400	
Direct effects: Middle SES—BP control	0.008	−0.006	0.024	0.320	
**High SES—BP control (mediating variable: WC)**					Suppressing effect
Total effects	0.084	0.054	0.121	<0.001	
Indirect effects: High SES—WC—BP control	−0.002	−0.003	−0.001	0.020	
Direct effects: High SES—BP control	0.086	−0.056	0.123	<0.001	

Notes: SES: socioeconomic status. BP: blood pressure. BMI: body mass index. WC: waist circumference. All analyses control for gender, age, marital status, smoking status, drinking status, and healthy physical activity, use of antihypertensive medication, numbers of antihypertensive medications, duration of hypertension at all paths.

## Data Availability

The data presented in this study are available on request from the corresponding author. The data are not publicly available due to privacy protection and confidentiality policies.

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
