# Peer review of "Mediation Effect of Obesity on the Association of Socioeconomic Status with Blood Pressure in the Elderly Hypertensive Population"

_nutrients, 2024, doi:10.3390/nu16152401_

Round 1

Reviewer 1 Report

Comments and Suggestions for Authors

Nutrients - 3065461 

Type of manuscript: Article
Title: Mediation effect of obesity on the association of socioeconomic status

with blood pressure in the elderly hypertensive population
Authors: Saiyi Wang, Yudong Miao, Yi-fei FENG, Li-pei ZHAO, Xiaoman Wu, Shiyu Jia, Weijia Zhao, Clifford Silver Tarimo, Yibo Zuo, Xing-hong GUO, Ming-ze, MA, Jian Wu *
Nutrition and Obesity

Reviewer’s comments:

The authors studied 17,234 Chinese hypertensive elderly people (mean age 73.4 years; 57.4% female) and aimed to evaluate the effects of socio-economic status (measured through educational level, family income and occupation) on blood pressure control and assessed the effects of BMI and waist circumference.

1.        Please make it clear to the reader whether this was a prospective or retrospective study.

2.        Please provide more information on how you recruited the study participants. You state on page 2 line 75 that you recruited them from the National Basic Public Health Program. How does that work?

3.        Please explain why the BMI is less than 25 kg/m2 on average? I know there is a difference between the definitions of the ranges of BMI in Caucasians vs Chinese/Asian people (normal – overweight – obese). This has not been addressed in the manuscript. How can you explain you did a study of the effect of obesity, if you do not have the ‘obese’ group included?

4.        The presentation of figure 1 should include the normal range of the BMI for low weight/normal weight/ overweight and obesity (or it should be mentioned in the main text). Why are data of ‘waist circumference’ not included in figure 1.

5.        Figure 1 – please correct the spelling of PREVALENCE; explain why you call it Ptrend and not P-value?

6.        Figure 2 – please correct: FARMER (not famer); NO-FARMER (not no-famer) – please check and correct this in the whole manuscript; see e.g. p2 line 93; p. 14 – suppl Table 6; p.15 suppl Figure 1.

7.        Please present the data in your tables in a more readable format.

a.        Please put under each table, what exactly the data are: mean (SD) or % or medium….?

b.       Please explain under each table the abbreviations used in full.

c.        Table 2: Model 1, 2, 3, 4 – all data should fit on one line; again why Ptrend?; put a space between Model and the number.

d.       Suppl Table 1: The way you have presented the data in columns (AIC, BIC, aBIC) are impossible to understand. Please redo this table.

8.        Please add a space between the data, i.e., p 1, line 28: 0.912(0.854,0.974) è 0.912 (0.854, 0.974). Check and correct this throughout the whole manuscript. See data in Suppl tables 3-4-5-6).

9.        Please add ‘years’ after 73.4 on page 1 – line 24.

10.   Please change ‘were differ’ into ‘differed’ on page 2 – line 49.

11.   Please check (and delete one) of the P<0.05 indicated statistical significance (see p 4 lines 144-145).

12.   Please rephrase on p 9 – line 222: The Chinese study discovered that in plateau area, the lower SES was, the higher the level of BP was. Please ask help to correct English grammar.

13.   Conclusion p 10 line 276: ‘SES was associated with the risk of high BP’. Please rephrase the first sentence. What you want to say is that your study on an elderly hypertensive population with high SES showed better results for blood pressure controle …. and add the impact of BMI/WC … - Please rephrase the whole conclusion as it does not sound good at this moment. Ask help to correct the English grammar.

14.   References: see #18: @ WHO (p 11 line 343) should follow on line 342).

Comments on the Quality of English Language

Suggest to ask a native English-speaking person to review the manuscript. It will help improve the text.

Author Response

Reviewer #1

The authors studied 17,234 Chinese hypertensive elderly people (mean age 73.4 years; 57.4% female) and aimed to evaluate the effects of socio-economic status (measured through educational level, family income and occupation) on blood pressure control and assessed the effects of BMI and waist circumference. _____________________________________________________________________

Point 1. Please make it clear to the reader whether this was a prospective or retrospective study.

Response: Thank you for your consideration. Our study was a cross-sectional study conducted countywide in 2023. Our study design is a prospective study with planned follow-up every two years, and this study is a baseline investigation. We have redescribed it in the methods section," The study was a county-wide cross-sectional survey of Chinese hypertensive population conducted between July 1 and August 31, 2023 in Jia County, Henan Province." (Page 2, lines 80-81).

_____________________________________________________________________

Point 2. Please provide more information on how you recruited the study participants. You state on page 2 line 75 that you recruited them from the National Basic Public Health Program. How does that work?

Response: Thank you for your suggestion. Study participants were recruited from the National Basic Public Health Program, a government initiative in China aimed at improving public health through comprehensive and accessible health services. This program maintains extensive health records and provides routine health services to residents, including hypertension management and monitoring.

Specifically, we collaborated closely with local health authorities in Jia County to access their databases and identify eligible hypertensive patients. Our inclusion criteria were patients aged 65 years or older, diagnosed with hypertension, who agreed to participate in the study and signed an informed consent form. The exclusion criteria are 1) Patients with mental illness and dementia. 2) Patients with secondary hypertension. 3) Patients who were disabled or seriously disabled and unable to complete the questionnaire and physical examination. Based on these criteria, the eligible participants were recruited. This recruitment approach ensured a representative sample of the hypertensive population in Jia County, facilitating a robust cross-sectional analysis.

Therefore, we made corresponding changes in the manuscript. “The subjects were hypertensive patients aged 65 years or older, enrolled through the National Basic Public Health Service Program. They had been diagnosed with hypertension, agreed to participate in the study, and signed an informed consent form. Subjects with mental illness, secondary hypertension, and those with disabilities or severe disabilities who were unable to complete the questionnaires and physical examinations were excluded. A total of 18,963 hypertensive patients were investigated.” (Page 2, lines 81-87)

_____________________________________________________________________

Point 3. Please explain why the BMI is less than 25 kg/m2 on average? I know there is a difference between the definitions of the ranges of BMI in Caucasians vs Chinese/Asian people (normal – overweight – obese). This has not been addressed in the manuscript. How can you explain you did a study of the effect of obesity, if you do not have the ‘obese’ group included?

ResponseThank you for your consideration. In our previous analysis, BMI in Table 1 was presented as a continuous variable. Since it did not follow a normal distribution, it was expressed using the interquartile range P50 (P25, P75). Therefore, the values “24.6, 24.5, 24.7” in Table 1 are the median, not the mean values.

As you may have considered, there is a difference in BMI in Caucasians vs Chinese/Asian people (normal-overweight-obese). However, since our study population is Chinese, our BMI categories are defined according to the Chinese Adult Overweight and Obesity Prevention and Control guidelines. We have added a description to the methods section: “The guidelines specify four BMI categories: underweight (BMI < 18.5 kg/m2), normal weight (18.5 kg/m2≤ BMI < 24.0 kg/m2), overweight (24.0 kg/m2≤ BMI < 28.0 kg/m2), and obese (BMI ≥ 28.0 kg/m2)[19].” (Page 4, lines 146-147).

In our previous analysis, BMI was studied and analyzed as a continuous variable (including mediation analysis). Only in the subgroup analysis (Figure 1) did we presented the relationship between SES and BP control across different BMI groups. Therefore, we did not present four BMI groups in the manuscript, but our population included the obese group (N=3052 (17.7%), as well as the underweight, normal weight, and overweight groups, with respective counts of 539, 6870, and 6773. Our obesity rate is consistent with the 14.1% reported by the Chinese Center for Disease Control and Prevention, which is within the expected range[19].

  1. China Obesity working group Guidelines for prevention and control of overweight and obesity in Chinese adults (excerpt) [J] Acta Nutrimenta Sin, 2004,26(1):1-4, doi:10.3321/j.issn:0512-7955.2004.01.001.

_____________________________________________________________________

Point 4. The presentation of figure 1 should include the normal range of the BMI for low weight/normal weight/ overweight and obesity (or it should be mentioned in the main text). Why are data of ‘waist circumference’ not included in figure 1.

ResponseThank you for your suggestion. We have added a description to the methods section: “The Guidelines specify four BMI categories: underweight (BMI < 18.5 kg/m2), normal weight (18.5 kg/m2≤ BMI < 24.0 kg/m2), overweight (24.0 kg/m2≤ BMI < 28.0 kg/m2), and obese (BMI ≥ 28.0 kg/m2)[19].” (Page 3, lines 142-144). In addition, a subgroup analysis of waist circumference was omitted in previous studies due to my oversight, we have now included a subgroup analysis of different waist circumference groups in Figure 1. The definition of waist circumference was also added in the method part: “Central obesity: male WC ≥ 90 cm, female WC ≥ 85 cm”. (Page 4, lines 147-148).

_____________________________________________________________________

Point 5. Figure 1 – please correct the spelling of PREVALENCE; explain why you call it Ptrend and not P-value?

ResponseThank you for your suggestion. We have corrected the spelling error in Figure 1 and changed “Ptrend ” to “P-value”.

_____________________________________________________________________

Ponit 6. Figure 2 – please correct: FARMER (not famer); NO-FARMER (not no-famer) – please check and correct this in the whole manuscript; see e.g. p2 line 93; p. 14 – suppl Table 6; p.15 suppl Figure 1.

ResponseThank you for your suggestion. We have modified the spelling error of “Farmer” and “Non-farmer” in Figure 2. Same errors throughout the text have also been proofread and modified.

Figure 2. Heatmaps of the associations between education, income, occupation, and blood pressure. Data are OR (95% CI). All models were adjusted for age, gender, marital status, drinking status, smoking status, healthy physical activity, the use of antihypertensive medication, numbers of antihypertensive medications, duration of hypertension, BMI, and WC. The colors are presented in a gradient, with a darker color shade indicating a stronger association. *: P<0.05.

_____________________________________________________________________

Point 7. Please present the data in your tables in a more readable format.

  1. Please put under each table, what exactly the data are: mean (SD) or % or medium….?
  2. Please explain under each table the abbreviations used in full.
  3. Table 2: Model 1, 2, 3, 4 – all data should fit on one line; again why Ptrend?; put a space between Model and the number.
  4. Suppl Table 1: The way you have presented the data in columns (AIC, BIC, aBIC) are impossible to understand. Please redo this table.

Response:Thank you for your consideration.

Regarding point a, we have specified the meaning of the data under the table. “Notes: Data were reported as a percentage (%) for categorical variables, and the interquartile range P50 (P25, P75) for continuous variables that do not conform to a normal distribution. SBP: systolic blood pressure. DBP: diastolic blood pressure.” (Page 7, lines 203-205; Page 13, lines 448-450; Page 14, lines 452-454; Page 14, lines 456-458; Page 15, lines 460-462).

Regarding point b, we have explained the abbreviations in the table notes. For example: “SBP: systolic blood pressure. DBP: diastolic blood pressure.” (Page 7, lines 203-205); “SES: socioeconomic status.” (Page 8, lines 245).

Regarding point c. Thank you for your careful review. We have added a space between “Model” and the number. (Page 8, Table 2) The reason for analyzing Ptrend is that we previously examined the impact of medium SES and high SES on blood pressure control compared with low SES. To investigate the change in blood pressure control when SES increases by one unit, we conducted a Cochran-Armitage trend test. Ptrend < 0.05 indicated a statistically significant trend. Additionally, the results reaffirmed that higher SES was associated with a better blood pressure control.

Regarding point d, we have modified Suppl Table 1 and added the corresponding explanation below the table (Page 12, lines 442-445). “Notes: AIC: Akaike information criterion; BIC: Bayesian information criterion; aBIC: adjusted Bayesian information criterion; LMR-LRT: Lo-Mendell-Rubin likelihood ratio test; BLRT, Bootstrapped likelihood ratio test. For LMR-LRT and BLRT, non-significant results suggest that fewer profiles are appropriate.

_____________________________________________________________________

Point 8. Please add a space between the data, i.e., p 1, line 28: 0.912(0.854,0.974) è 0.912 (0.854, 0.974). Check and correct this throughout the whole manuscript. See data in Suppl tables 3-4-5-6).

Response:Thank you for your suggestion. We have added a space to the whole manuscript and table, and the changes are highlighted in yellow. (See data in Table 1, Table 2, and Suppl Tables 3-4-5-6).

_____________________________________________________________________

Point 9. Please add ‘years’ after 73.4 on page 1 – line 24.

Response:Thank you for your suggestion. We have added the word “years” after “73.4” in the text. (Page 1, line 28).

_____________________________________________________________________

Point 10. Please change ‘were differ’ into ‘differed’ on page 2 – line 49.

Response:Thank you for your suggestion. We have corrected these minor errors in the revised manuscript (Page 2, line 56), and carefully reviewed the entire text to avoid similar issues.

_____________________________________________________________________

Point 11. Please check (and delete one) of the P<0.05 indicated statistical significance (see p 4 lines 144-145).

Response:Thank you for your suggestion. We have deleted this sentence – “And P values <0.05 indicated statistical significance.” (Page 4, line 175).

_____________________________________________________________________

Point 12. Please rephrase on p 9 – line 222: The Chinese study discovered that in plateau areas, the lower SES was, the higher the level of BP was. Please ask help to correct English grammar.

Response: Thank you for your careful reading of our manuscript. We have revised the grammar of this sentence: “The study from China found that in plateau regions, lower SES was associated with higher BP levels.” (Page 9, line 257).

_____________________________________________________________________

Point 13. Conclusion p 10 line 276: ‘SES was associated with the risk of high BP’. Please rephrase the first sentence. What you want to say is that your study on an elderly hypertensive population with high SES showed better results for blood pressure controle …. and add the impact of BMI/WC … - Please rephrase the whole conclusion as it does not sound good at this moment. Ask help to correct the English grammar.

Response: Thank you for your suggestion. We have revised the sentence: “In conclusion, our study found that SES was associated with the risk of high BP. Additionally, obesity indicators, such as BMI and WC, were identified as suppressive factors in the relationship between high SES and BP control. These findings highlighted the importance of not only to improve SES but also to address obesity effectively for better BP control.” (Page 10, lines 325-328).

_____________________________________________________________________

Point 14. References: see #18: @ WHO (p 11 line 343) should follow on line 342).

Response: Thanks for your careful reading of our manuscript. We have modified the format of reference [18] “WHO Guidelines on Physical Activity and Sedentary Behaviour. (2020). World Health Organization.” (Page 11, lines 390).

Reviewer 2 Report

Comments and Suggestions for Authors

The article's topic is interesting and the study was conducted on a wide population of hypertensive patients. However, several criticisms should be addressed:

- In the abstract, a better characterization of the study population should be shown. Who were these patients? Where did they come from and where were they evaluated? Are they all on anti-hypertensive treatment?

- In the abstract, it is not clear the analysis, which demonstrates how BMI/WC affect the relationship between SES and BP control, which is the central point of the study.

- The authors did not mention the anti-hypertensive treatment. This could affect the relationship found between SES and BP control.

- More details on the BP measurement should be reported in the methods section (manual or automatic oscillometric device, adequate cuff size ...). Further details should be specified regarding weight, height and WC measurement.

-  Figure 1 is not easily understandable. Why was it included in the methods instead of the results section? It shows the risk of BP control associated with socioeconomic status by several covariates, however it is not clear what the reference state is within the different variables examined. Please review the figure or the caption.

- In Supplementary Figure S2, a horizontal dotted line should be placed on the OR level equal to 1 to facilitate reading of the figure.

- Throughout the manuscript, rather than reporting the inverse associations referring to non-control, it would be easier to interpret and more correct to report the probability of being controlled, therefore reporting OR>1.

- The variable “take medication situation” is reported only in the adjusted models, however, it is not defined in the methods. It is an important point to clarify. The number and dosage of drugs taken by patients could greatly affect BP control and could also vary based on SES, given that some drugs could have a contribution to the payment or in any case more drugs often require more visits and checks. What information do the authors have available on this issue? At least the number of drugs taken should be taken into account.

- In the discussion, the authors should keep the association between SES and BP control clearly distinct from the association between SES and the risk of hypertension onset. If, on the one hand, it is true that a higher SES is associated with better CP control and this is at least partly mediated by better adiposity indices (as well as probably better access to health care), on the other hand, it is also true that a lifestyle with sedentary and hypernutrition, typical of high SES, could predispose to the onset of hypertension.

- Furthermore, in the discussion, the authors should better explore the mechanisms which link adiposity and high BP (PMID: 38654832).

- The conclusions also need to be reviewed. The authors did not study the risk of hypertension, but BP control in hypertensive patients. 

Comments on the Quality of English Language

No particular criticism found.

Author Response

Reviewer #2

The article's topic is interesting and the study was conducted on a wide population of hypertensive patients. However, several criticisms should be addressed:

Point 1. In the abstract, a better characterization of the study population should be shown. Who were these patients? Where did they come from and where were they evaluated? Are they all on anti-hypertensive treatment?

Response: Thank you for your careful reading of our manuscript. Our study population comprised hypertension patients from Jia County in Henan Province who are enrolled in the National Basic Public Health Service Program. The patients were diagnosed with hypertension by a secondary or higher medical institution and subsequently enrolled in the program. In 2023, we conducted a questionnaire survey and physical examination of 18963 hypertensive patients across the entire county. During the survey, we discovered that not all patients were receiving high blood pressure medication.

In the abstract section of our revised manuscript, we have provided more details regarding the study population: “The study was conducted in Jia County, Henan Province, China, from July 1 to August 31, 2023. The 18963 hypertensive population over 65 years old who were included in the National Basic Public Health Service Program were investigated.” (Page 1, lines 18-20).

_____________________________________________________________________

Point 2. In the abstract, it is not clear the analysis, which demonstrates how BMI/WC affect the relationship between SES and BP control, which is the central point of the study.

ResponseThank you for your consideration. Our study revealed that BMI and waist circumference (WC) play a suppressive role in the relationship between high socioeconomic status (SES) and blood pressure (BP) control. While elderly hypertensive individuals with higher SES demonstrated better BP control, increases in BMI and WC undermined this beneficial relationship. Therefore, implementing obesity prevention strategies is essential to maintain the beneficial association between high SES and BP control. We have included the following additions in the manuscript: “Compared with low SES, our findings further suggested that BMI (Indirect effects: 95% CIs: 0.002—0.011; P=0.007) and WC (Indirect effects95% CIs0.001—0.009; P=0.025) play a suppressing role in the association between high SES and BP control.”(Page 1, lines 32-34). “This indicated that despite the better BP control observed in elderly hypertensive populations with high SES, BMI and WC may undermine this beneficial relationship.” (Page 1, lines 37-39).

_____________________________________________________________________

Point 3. The authors did not mention the anti-hypertensive treatment. This could affect the relationship found between SES and BP control.

ResponseThank you for your consideration. In our initial submission of the manuscript, we adjusted for the use of antihypertensive medications among the hypertensive population in the analysis of socioeconomic status (SES) and blood pressure (BP) control. To further mitigate the influence of confounding factors, we incorporated the specific numbers of antihypertensive medications taken as a moderating variable in our model. Definitions for “use of antihypertensive medication” and “numbers of antihypertensive medications” are provided in method 2.4. Covariate Assessment section. “Participants provided information on their use of antihypertensive medication, including whether they were currently taking it as well as the specific numbers of medication they used)”. (Page 3, lines 134-136).

_____________________________________________________________________

Point 4. More details on the BP measurement should be reported in the methods section (manual or automatic oscillometric device, adequate cuff size ...). Further details should be specified regarding weight, height and WC measurement.

ResponseThank you for your consideration. We have provided detailed information on blood pressure, height, weight, and waist measurements in the manuscript.

Blood Pressure Measurement: “We measured BP levels three times using a standard Omicron electronic sphygmomanometer. Participants were instructed to sit still for five minutes before the first measurement. During the measurements, participants placed their left hand palm up, and a trained primary care worker positioned the armband around the upper arm. The bottom of the armband was positioned 1-2 cm above the inner elbow joint, ensuring it did not cover the joint. The measurement error did not exceed 10 mmHg, and the mean of the three measurements was used to define the systolic and diastolic BP levels.” (Page 3, lines 113-118).

Weight, Height, and Waist Circumference Measurement: “Participants were instructed to remove any accessories and outer layers of clothing such as hats, coats or jackets during height and weight measurements. Weight and WC measurements were taken in a fasting state. WC was measured at a level 1 cm above the upper edge of the navel. Each measurement was repeated three times, and the average value was calculated. (Page 3, lines 137-144).

_____________________________________________________________________

Point 5. Figure 1 is not easily understandable. Why was it included in the methods instead of the results section? It shows the risk of BP control associated with socioeconomic status by several covariates, however it is not clear what the reference state is within the different variables examined. Please review the figure or the caption.

ResponseThank you for your consideration. Figure 1 should appear in the results section, and we have deleted it in the methods section. In section “3.2 Association between SES and BP control”, we initially analyzed the relationship between socioeconomic status (SES) and blood pressure (BP) control, finding that middle and high SES were associated with improved BP control compared to low SES. To confirm the robustness of these findings, we conducted subgroup analyses. In our initial manuscript submission, we omitted the control group and subsequently revised Figure 1, changing “Ptrend” to “P” for clarity.

_____________________________________________________________________

Point 6. In Supplementary Figure S2, a horizontal dotted line should be placed on the OR level equal to 1 to facilitate reading of the figure.

Response: Thank you for your suggestion. In our initial manuscript submission, we ignored the line description of “OR=1” in Supplementary Figure S2. Therefore, we have revised Supplementary Figure S2 accordingly (See Page 16, line 464).

Supplementary Figure 2. Association of educational level, income, occupation with BP control. a): the association of educational level and BP control; b): the association of income and BP control; c): the association of occupation and BP control. Notes: The model adjusted for gender, age, smoking status, drinking status, healthy physical activity, marital status, and antihypertensive medication use situation, numbers of antihypertensive medications, hypertension duration. *: P<0.05.

_____________________________________________________________________

Point 7. Throughout the manuscript, rather than reporting the inverse associations referring to non-control, it would be easier to interpret and more correct to report the probability of being controlled, therefore reporting OR>1.

Response: Thank you for your suggestion. We have revised the results in the manuscript to report the probability of being controlled. The description of OR > 1 has been presented accordingly.

_____________________________________________________________________

Point 8. The variable “take medication situation” is reported only in the adjusted models, however, it is not defined in the methods. It is an important point to clarify. The number and dosage of drugs taken by patients could greatly affect BP control and could also vary based on SES, given that some drugs could have a contribution to the payment or in any case more drugs often require more visits and checks. What information do the authors have available on this issue? At least the number of drugs taken should be taken into account.

Response: Thank you for your suggestion. We supplemented the description of the antihypertensive medication use situation in the method, and added the variable of numbers of antihypertensive medications for subsequent regression models. “Participants provided information on their use of antihypertensive medication, including whether they were currently taking it as well as the specific numbers of medication they used))”. (Page 3, lines 133-135).

_____________________________________________________________________

Point 9. In the discussion, the authors should keep the association between SES and BP control clearly distinct from the association between SES and the risk of hypertension onset. If, on the one hand, it is true that a higher SES is associated with better CP control and this is at least partly mediated by better adiposity indices (as well as probably better access to health care), on the other hand, it is also true that a lifestyle with sedentary and hypernutrition, typical of high SES, could predispose to the onset of hypertension.

Response: Thank you for your suggestion. Indeed, studies in various regions have indicated that individuals with higher socioeconomic status (SES) may be more susceptible to developing hypertension due to lifestyle factors and other habits. Perhaps the differences in results could be due to differences in race, study design, living conditions, and the level of economic development of the population, which we also discussed in the revised manuscript. However, our study specifically examines the relationship between SES and blood pressure control in individuals with pre-existing hypertension, and our findings align with the majority of existing research.

In the discussion, we have shed new light on the relationship between SES and blood pressure control: “Research has shown that in developing countries, the adoption of a more Western lifestyle among high-SES individuals often involves increased consumption of fats, oils, and animal-based foods compared to other population groups [25]. Nevertheless, extensive evidence underscores that low SES is associated with adverse health outcomes, including heightened diabetes risk, increased rates of cardiovascular disease mortality and morbidity, overall mortality, and reduced life expectancy [26,27].”

  1. Leng, B.; Jin, Y.; Li, G.; Chen, L.; Jin, N. Socioeconomic status and hypertension: a meta-analysis. Journal of hypertension 2015, 33, 221-229, doi:10.1097/hjh.0000000000000428.
  2. Stringhini, S.; Carmeli, C.; Jokela, M.; Avendaño, M.; Muennig, P.; Guida, F.; Ricceri, F.; d'Errico, A.; Barros, H.; Bochud, M., et al. Socioeconomic status and the 25 × 25 risk factors as determinants of premature mortality: a multicohort study and meta-analysis of 1·7 million men and women. Lancet (London, England) 2017, 389, 1229-1237, doi:10.1016/s0140-6736(16)32380-7.
  3. Zhang, Y.B.; Chen, C.; Pan, X.F.; Guo, J.; Li, Y.; Franco, O.H.; Liu, G.; Pan, A. Associations of healthy lifestyle and socioeconomic status with mortality and incident cardiovascular disease: two prospective cohort studies. BMJ (Clinical research ed.) 2021, 373, n604, doi:10.1136/bmj.n604.

_____________________________________________________________________

Point 10. Furthermore, in the discussion, the authors should better explore the mechanisms which link adiposity and high BP (PMID: 38654832).

Response: Thank you for your suggestion. In the discussion, we added a description of the mechanisms linking adiposity and high blood pressure: “Previous studies have demonstrated that excess adipose tissue in obese hypertensive patients directly triggers chronic overactivation of the sympathetic nervous system (SNS), leading to increased heart rate and vasoconstriction, thereby elevating blood pressure [35]. Additionally, obesity often activates the renin-angiotensin-aldosterone system (RAAS), resulting in heightened levels of renin and angiotensin. These substances promote vasoconstriction and enhance sodium and water retention, further contributing to elevated blood pressure [36]. Reduced levels of adiponectin in obese populations diminish its protective effects on blood vessels, thereby increasing hypertension risk. Furthermore, leptin, a hormone secreted by fat cells, can elevate blood pressure by stimulating the sympathetic nervous system [37].” (Page 9-10, lines 299-309).

  1. Grassi, G.; Dell'Oro, R.; Facchini, A.; Quarti Trevano, F.; Bolla, G.B.; Mancia, G. Effect of central and peripheral body fat distribution on sympathetic and baroreflex function in obese normotensives. Journal of hypertension 2004, 22, 2363-2369, doi:10.1097/00004872-200412000-00019.
  2. Oikonomou, E.K.; Antoniades, C. The role of adipose tissue in cardiovascular health and disease. Nature reviews. Cardiology 2019, 16, 83-99, doi:10.1038/s41569-018-0097-6.
  3. Sarzani, R.; Landolfo, M.; Di Pentima, C.; Ortensi, B.; Falcioni, P.; Sabbatini, L.; Massacesi, A.; Rampino, I.; Spannella, F.; Giulietti, F. Adipocentric origin of the common cardiometabolic complications of obesity in the young up to the very old: pathophysiology and new therapeutic opportunities. Frontiers in medicine 2024, 11, 1365183, doi:10.3389/fmed.2024.1365183.

_____________________________________________________________________

Point 11. The conclusions also need to be reviewed. The authors did not study the risk of hypertension, but BP control in hypertensive patients.

Response: Thank you for your suggestion. The results suggest that we should not only enhance SES but also address obesity to improve BP control in hypertensive patients effectively. Therefore, we have revised the conclusions section description: “In conclusion, our study found that SES was associated with the risk of high BP. Additionally, obesity indicators, such as BMI and WC, were identified as suppressive factors in the relationship between high SES and BP control. These findings highlighted the importance of not only to improve SES but also to address obesity effectively for better BP control.” (Page 10, lines 325-328).

Round 2

Reviewer 2 Report

Comments and Suggestions for Authors

The authors responded well to the criticisms. Only a minor comment remains:

- The figure legend of Figure 1 should be amended to “Adjusted odds ratio (ORs) of blood pressure control associated with socioeconomic status according to the subpopulation” or something similar. Moreover, the authors should report that “Low SES is the reference group”. 

Comments on the Quality of English Language

None. 

Author Response

Point 1. The figure legend of Figure 1 should be amended to “Adjusted odds ratio (ORs) of blood pressure control associated with socioeconomic status according to the subpopulation” or something similar. Moreover, the authors should report that “Low SES is the reference group”. 

Response: Thank you for your suggestion. We have modified the legend in Figure 1: “Adjusted odds ratio (ORs) of blood pressure control associated with socioeconomic status according to the subpopulation (Low SES is the reference group)”. (Page 5, lines 169-170). Similarly, we have also added a description under Table 2: “Low SES is the reference group”. (Page 8, line 215).